# Inharmonicity enhances brain signals of attentional capture and auditory stream segregation
Krzysztof Basiński [1] ✉, Alexandre Celma-Miralles [2], David R. Quiroga-Martinez [2,3] & Peter Vuust[2]

Harmonicity is an important feature for auditory perception. However, the neural substrates of processing inharmonic sounds remain unclear. Here, we systematically manipulated the harmonicity of sounds by introducing random jittering to their frequencies. Using electroencephalography, we studied the effect of inharmonicity on markers of auditory prediction errors — mismatch negativity (MMN) and P3a — in a roving oddball paradigm. Inharmonic sounds with a constant jittering pattern generated similar MMN and stronger P3a responses than harmonic sounds. In contrast, MMN responses became undetectable when the jittering pattern changed between consecutive sounds, suggesting that prediction errors are weighted by sequential but not spectral uncertainty. Interestingly, inharmonic sounds generated an object-related negativity, a response associated with the segregation of auditory objects. Our results suggest that inharmonicity induces the segregation of the auditory scene into different streams, captures attention, and gives rise to specific neural processes that are independent from the predictive mechanisms underlying sequential deviance detection.

Harmonic sounds consist of acoustic waves containing frequencies that are integer multiples of a common fundamental (F0). Many ecologically salient sounds, such as human voices, animal calls or music, are highly harmonic. Conversely, if the acoustical wave consists of frequencies that are not integer multiples of a common fundamental, it is said to be inharmonic. These sounds are often described as noises, sizzles, pops or rattles[1]. Harmonicity has been suggested to underlie auditory scene analysis[2] and help separate the source of relevant sounds from background noise[3], forming a basic organizational principle of the auditory cortex[4–6]. In language, violations of harmonicity impair the ability to understand concurrent speech and cause listeners to lose track of target talkers[7]. In music, inharmonic sounds can impair pitch-related abilities such as interval detection[8]. Similarly, inharmonicity disrupts the ability to compare pitch across time delays, which suggests a role in memory encoding[9].

Harmonicity can be understood in the light of predictive processing theories, which posit that perception relies on the brain forming top-down predictions about the incoming stimuli and their causes[10–17]. These predictive processing accounts assume that predictions stem from an internal, generative model that is constantly updated by statistical regularities in the incoming sensory information. An "error" or "surprise" response (i.e., *prediction error*) occurs when predictions do not fit the sensory input, which can be used to update the contents of the generative

model[12,18]. According to this theory, prediction errors are weighted by uncertainty (or precision) of the context, adjusting the relevance given to sensory inputs, allowing perceptual inference under uncertainty[19–21]. Therefore, if a signal is imprecise or unreliable (as in, for example, seeing in dark conditions or hearing in noisy environments), any prediction errors arising from it would be down-weighted and less likely to influence future predictions. While precision-weighting in the auditory domain has been explored empirically (see ref. 13 for a comprehensive review), the details of this process remain unclear.

Here, we hypothesize that harmonicity might be one of the relevant auditory features (among others, such as duration or intensity) that drive precision-weighting of prediction errors. Harmonic sounds have lower information content (*lower entropy*[22]) than inharmonic sounds, as the spectrum of an ideally harmonic sound will consist only of integer multiples of F0. Thus, many aspects of the sound can be reliably described using only one piece of information: the F0. Conversely, inharmonic sounds have higher information content (*higher entropy*). Consequently, any prediction errors produced by inharmonic sounds should have lower precision and would therefore be less likely to influence future expectations[23]. In the case of inharmonic sounds, precision-weighting might affect prediction errors not only by weakening pitch percepts but, more generally, by down-weighting sensory evidence about the spectral content of sounds.

[1]Auditory Neuroscience Laboratory, Department of Psychology, Medical University of Gdańsk, Gdańsk, Poland. [2]Center for Music in the Brain, Department of Clinical Medicine, Aarhus University & The Royal Academy of Music Aarhus/Aalborg, Aarhus, Denmark. [3]Department of Psychology, University of Copenhagen, Copenhagen, Denmark. ✉e-mail: k.basinski@gumed.edu.pl

Cortical responses associated with prediction errors can be quantified in event-related potentials (ERPs) using electroencephalography (EEG). Mismatch negativity (MMN) is a widely studied response to a deviation in an otherwise repetitive train of sensory stimuli[24]. It is a well-established electrophysiological trace of neural activity associated with precision-weighted prediction errors in the cortical auditory system[12,25-28]. Another neural response relevant to predictive coding is the P3 component, elicited when individuals shift attention (i.e., P3a) or consciously detect deviant stimuli (P3b[29]). In this study, we hypothesize that if harmonic sounds index precision as proposed by predictive processing theories, they should produce larger MMN and P3 responses to pitch deviants in comparison to inharmonic sounds.

Consistent with this idea, we previously showed that MMN responses were more prominent in harmonic piano tones than inharmonic hi-hat cymbal sounds[30]. However, due to the naturalistic nature of the stimuli, we could not entirely rule out if other factors, such as the presence of a stable pitch percept, rise and decay of the sound envelope or other spectral differences, played a role in the modulation of mismatch responses. To address this issue, here we present an EEG experiment using a passive roving oddball paradigm that carefully controlled the harmonic structure of sounds to isolate and measure the effect of harmonicity on auditory prediction errors. We recorded mismatch responses to sounds in three conditions: harmonic, inharmonic and changing. In the *harmonic* condition, sounds were regular harmonic complex tones. In the *inharmonic* condition, we introduced inharmonicity by jittering the frequencies above F0 and applied the same pattern of jitters to all tones in the sequence, thereby increasing the spectral uncertainty of the sounds. Note that the *inharmonic* condition introduces the same kind of uncertainty to the spectrum of each sound. However, MMN responses are also affected by the sequential uncertainty of the sound stream (i.e, what sound follows next). To investigate whether these two types of uncertainty interact, we introduced a *changing* condition in which a different jitter pattern was assigned to each individual sound. This increased sequential uncertainty and made predictions of subsequent sounds harder. We hypothesize here that the amplitudes of MMN and P3a would be affected by the spectral and sequential uncertainty of the tones. This would be shown as reduced prediction errors in inharmonic sounds, especially when the spectral content is repetitively jittered in the sequence.

## Results
### Paradigm outline
We used a version of the roving oddball paradigm to generate mismatch responses[31,32]. In a typical roving paradigm, several sounds are presented at a specific frequency, followed by a set of sounds at a different, randomly chosen frequency, which in turn is followed by another set, and so on. The sound immediately after the frequency shift is the deviant sound, but it eventually becomes the standard after a few repetitions. The participant listens to the sounds passively while watching a silent movie. In this study, instead of pure tones, we used (in)harmonic complex tones with the roving reflected in changes to the fundamental frequency of the complex. We used the roving paradigm because it ensures that the standards and deviants have the same physical properties (i.e., deviance is induced by the context of the sequence). It also allows for systematic investigation of the effect of the amount of frequency change on the mismatch response by taking advantage of the random changes to frequencies. Finally, the roving paradigm allows for investigating if mismatch responses are sustained after the first deviant in the sequence by analyzing the second and third-order deviants (shades of red in Fig. 1A, D, G).

In order to investigate the effect of harmonicity on mismatch responses, we presented sounds in three conditions. In the *harmonic* condition, all sounds were harmonic complex tones (consisting of F0 and its integer multiples, Fig. 1A–C. In the *inharmonic* condition, we introduced a random jittering pattern to each frequency above the F0 (Fig. 1D–F). Here, the same jittering pattern was applied to each sound within a set (Fig. 1D). In the *changing* condition, a different jittering pattern was applied to each sound. Crucially, in each of the conditions, the F0 remained unchanged by

the jittering. We assumed that keeping the lowest frequency constant would induce the perception of a fundamental (F0) pitch strong enough to form a predictive model based on this percept. Approximate entropy, a measure of the amount of information contained in a time series, was on average lower for harmonic ($M = 0.02$, SD = 0.01) than for inharmonic sounds ($M = 0.19$, SD = 0.01), and this difference was consistent irrespective of the F0 (Fig. 1H).

### Presence of MMN and P3
First, we investigated if the three conditions produced significant mismatch responses by testing the differences between standard and deviant responses using a mass-univariate approach. This allows for comparing ERPs without assumptions about spatio-temporal "windows" of activity[33]. We used cluster-based permutations, a non-parametric technique that tests for differences between conditions while controlling for multiple comparisons[34]. In the *harmonic* condition (Fig. 2A), we found differences between standards and deviants corresponding to a cluster at 87–219 ms (26/30 sensors, $p = 0.0002$). Topographically, this mismatch response started as a fronto-central negativity at 120–180 ms, typical of an MMN. A cluster at 216–286 ms, reflecting a frontocentral positivity typical of a P3a response, was marginally significant (16/30 sensors, $p = 0.076$). We also found an additional negative cluster at 329–450 ms (21/30 sensors, $p = 0.002$). In the *inharmonic* condition (Fig. 2B), we found differences between conditions corresponding to clusters in the data at 95–199 ms (23/30 sensors, $p = 0.002$) as well as at 198–359 ms (23/30 sensors, $p = 0.0001$). These overlapping clusters formed around typical latencies for MMN and P3a and had typical topographies. However, we found no differences between standards and deviants in the *changing* condition ($p > 0.05$ for all detected clusters), indicating that this condition did not produce any clear MMN or P3a responses (Fig. 2C and Table S1).

### Mass-univariate analysis of group differences
To investigate the differences between the three conditions, we used a cluster-based permutations approach, this time with an $F$-test. The test revealed differences between the conditions corresponding to clusters at 72–193 ms (25/30 sensors, $p = 0.016$) and at 211–345 ms (21/30 sensors, $p = 0.008$), indicating a main effect of condition (Fig. 3A). Next, we ran post-hoc pairwise comparisons between conditions using cluster-based $t$-tests (Fig. 3B). To account for multiple comparisons, we assumed a Bonferroni-corrected alpha level of $p = 0.05/3 = 0.0166$. We found significant differences between *harmonic* and *inharmonic* conditions in the P3a latency range (190-353 ms, a cluster comprising 22 sensors, $p = 0.010$) but not in the MMN range. Additionally, we found differences between *harmonic* and *changing* conditions for both MMN and P3a latency ranges (77–254 ms, 24/30 sensors, $p = 0.002$). A later cluster (363-450 ms) was not significant after multiple comparisons correction (20/30 sensors, $p = 0.033$). Finally, we found differences between *inharmonic* and *changing* conditions in the P3a range (200–358 ms, 24/30 sensors, $p = 0.001$). An earlier cluster (98–198 ms) corresponding to the MMN latencies was marginally significant (21/30 sensors, $p = 0.025$).

### Mean amplitude and peak latency
Next, in order to investigate the effect of harmonicity on the amplitudes and latencies of mismatch responses, we calculated participant-wise mean amplitudes and peak latencies in the latency ranges for both MMN and P3a (Fig. 4). For each of these measures we constructed linear mixed models with intercept and condition (*harmonic*, *inharmonic*, *changing*) as fixed effects and a random intercept for each participant (*m1*). We compared these models against null models that contained only the fixed intercept and a random intercept for each participant (*m0*).

For MMN mean amplitudes, *m1* performed significantly better than *m0* ($AIC_{m0} = 336.4$, $AIC_{m1} = 325.7$, $Chi^2(2) = 14.71$, $p = 0.0006$). Post-hoc comparisons of estimated marginal means revealed a significant difference between harmonic and changing conditions (contrast estimate = $-1.03$, SE = 0.27, $t(67.8) = -3.84$, $p = 0.0008$). Similarly, there was a significant

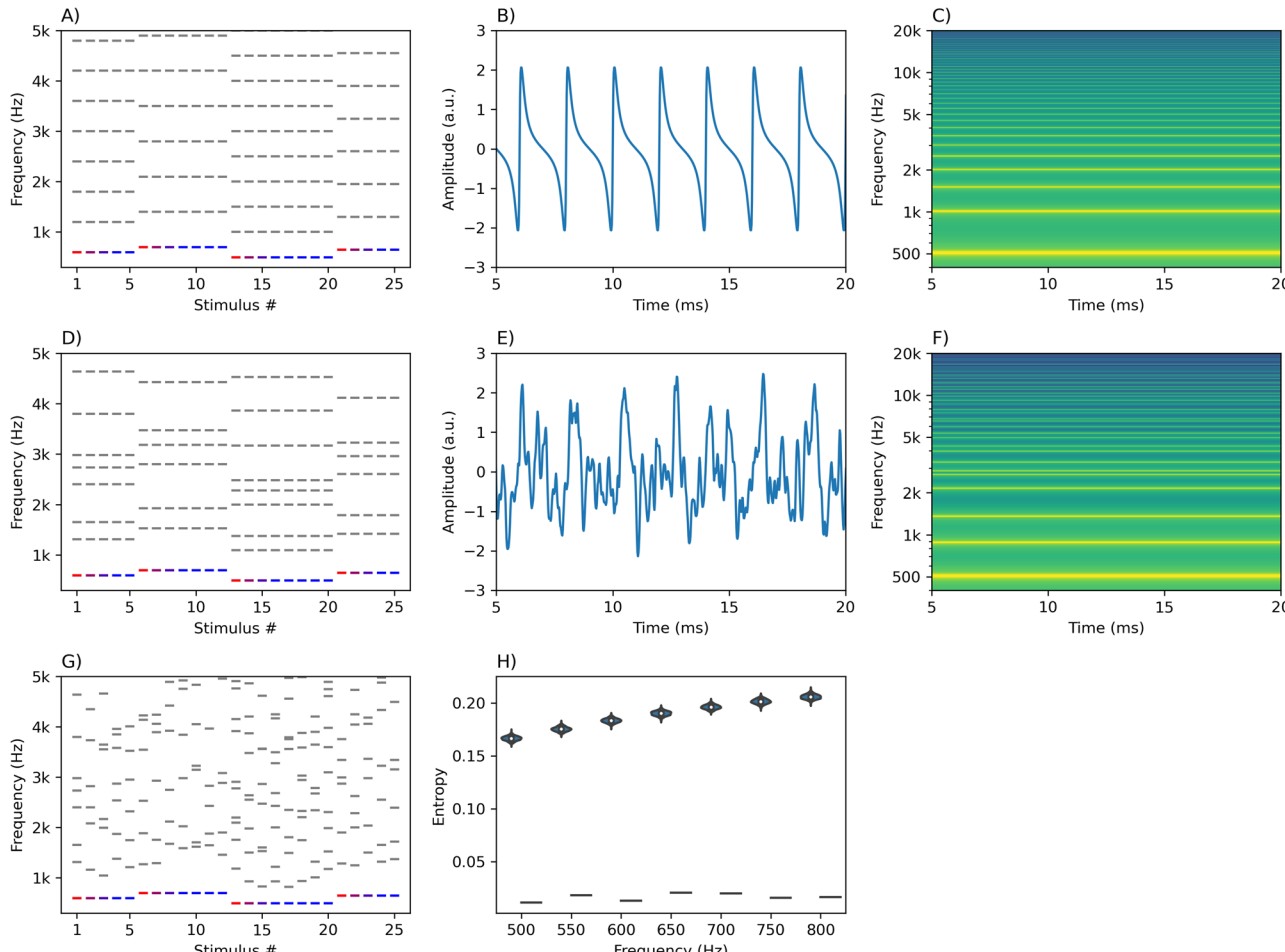

**Fig. 1 | Paradigm outline and stimulus characteristics. A** provides a symbolic representation of a roving sequence in the harmonic condition. The fundamental frequency is shown in blue or red, while the upper harmonics are shown in gray. Shades of red represent theoretical deviants, which progressively become standards (blue). **B** presents the waveform and **C** presents the spectrum of an example harmonic sound (F0 = 500 Hz). Similarly, **D–F** show sequence representation, waveform and spectrum for the inharmonic condition. Notice that while the distribution of the partials becomes uneven in the inharmonic condition, the pattern of jittering is carried through from one fundamental frequency to the next. Conversely, **G** shows the sequence representation for the changing condition, where a different jittering pattern is applied to each sound. **H** shows entropy calculations for harmonic (dashes) and inharmonic (violin plot distributions) sounds for each F0. Note that while we calculated entropies for 1000 different inharmonic sounds present in our sound pool, there was only one harmonic sound for each frequency (thus producing a single entropy value instead of a distribution).

---

difference between inharmonic and changing conditions (contrast estimate = −0.70, SE = 0.27, $t(67.8) = −2.57$, $p = 0.033$). The *harmonic–inharmonic* contrast was not statistically significant (contrast estimate = −0.34, SE = 0.27, $t(67.8) = −1.24$, $p = 0.43$). Taken together, these results suggest that there are no substantial differences in the MMN amplitude between *harmonic* and *inharmonic* sounds. However, the MMN is stronger (more negative) for both *harmonic* and *inharmonic* than for *changing* sounds.

For P3a mean amplitudes, *m1* also performed significantly better than *m0* ($AIC_{m0} = 458.8$, $AIC_{m1} = 444.0$, $Chi^2(2) = 18.80$, $p < 0.0001$). Post-hoc comparisons revealed significant differences between harmonic and inharmonic conditions (contrast estimate = -1.37, SE = 0.36, $t(68) = -3.85$, $p = 0.0007$) as well as between inharmonic and changing conditions (contrast estimate = 1.45, SE = 0.36, $t(68) = 4.07$, $p = 0.0003$). No significant differences were found for the harmonic-changing contrast (contrast estimate = 0.08, SE = 0.36, $t(68) = 0.23$, $p = 0.97$). These results suggest a significant effect of harmonicity on P3a amplitude, such that the P3a for inharmonic sounds is greater than for both harmonic and changing sounds.

For MMN peak latency measures, *m1* did not perform significantly better than *m0* ($AIC_{m0} = −390.4$, $AIC_{m1} = −388.0$, $Chi^2(2) = 1.59$, $p = 0.45$). The same was also the case for P3 peak latency ($AIC_{m0} = −331.7$, $AIC_{m1} = −330.2$ $Chi^2(2) = 2.49$, $p = 0.29$). This indicates that harmonicity does not substantially influence the latency of MMN and P3a mismatch responses.

### Object-related negativity

Beyond mismatch responses, the N1-P2 complex patterns in the *inharmonic* and *changing* condition differed noticeably from the *harmonic* condition (Fig. 2). A possible candidate for this type of response is the object-related negativity (ORN), an ERP component associated with a separation of concurrently presented sounds into distinct auditory objects[35–37]. Here, we investigated the possibility that inharmonic sounds gave rise to the ORN by contrasting the responses to *harmonic* vs. *inharmonic* and *harmonic* vs. *changing* sounds, for both standards and deviants. To this end, we performed cluster-based permutations with a *t*-test of differences between conditions (Fig. 5 and Table 1). We found differences corresponding to negative clusters in the data at latencies consistent with the ORN in *harmonic* vs. *inharmonic* contrasts (both for standards and for deviants) and in *harmonic* vs. *changing* contrasts (only for standards) (see Table S2 for a complete list of clusters detected in this analysis).

To investigate the relationship between the ORN and the deviance detection processes related to MMN and P3a, we extracted the P2 mean

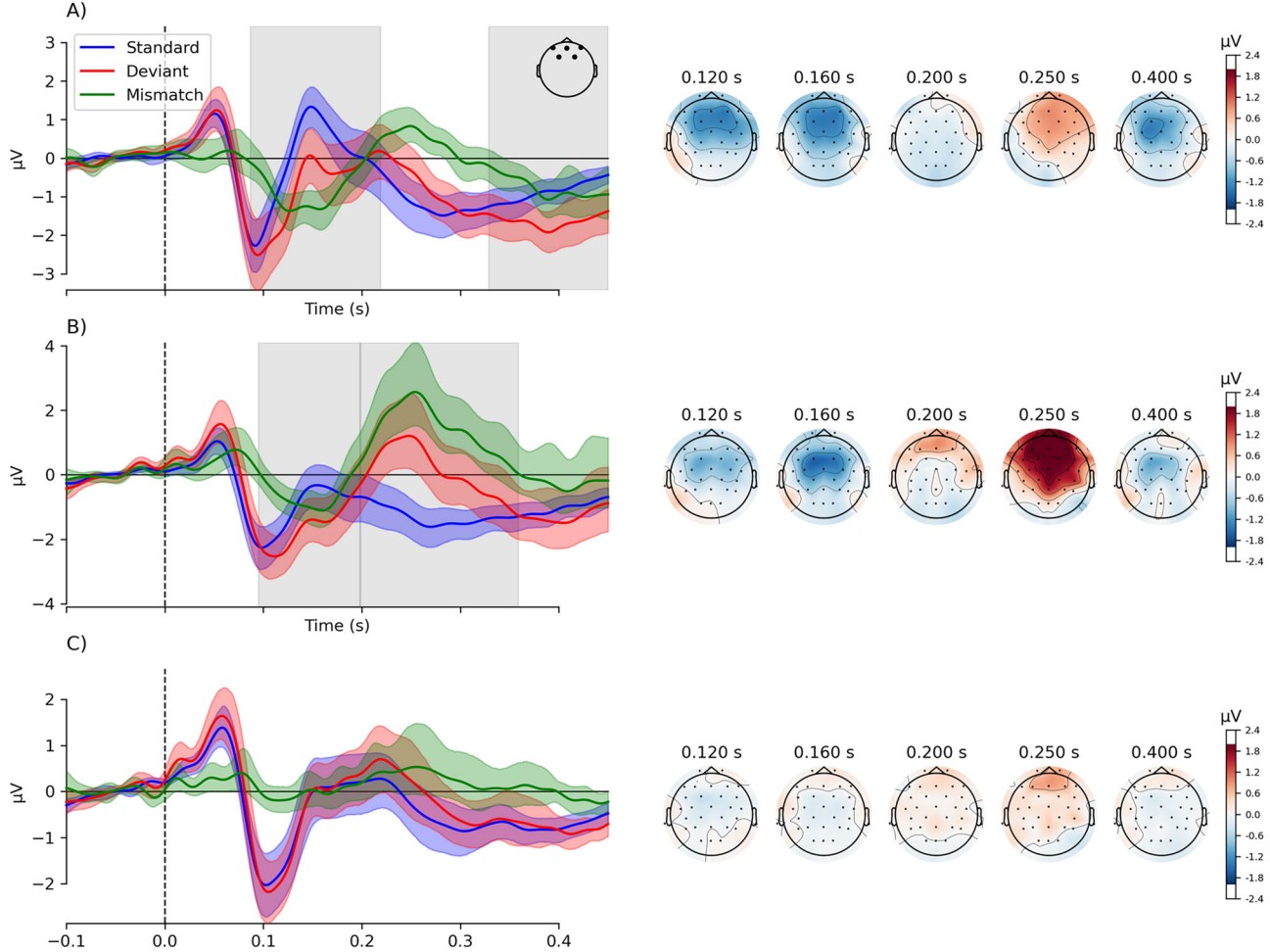

**Fig. 2 | Event-related potentials and their topographies.** Panels show responses in harmonic (**A**), inharmonic (**B**), and changing (**C**) conditions. Traces on the left show grand-average responses to standards (blue) and deviants (red), and the difference waves (green) for frontocentral channels (F3, Fz, F4, FC1, FC2). Color-shaded areas represent 95% confidence intervals, and gray shades represent significant clusters in the mass-univariate analysis. Scalp topographies present the difference wave signal strengths at chosen latencies.

amplitudes of both standards and deviants in each condition, averaged in a window around a global grand-average P2 peak latency (141 ms). We constructed a linear mixed model to predict P2 mean amplitude with *condition*, *deviance* (standard/deviant) and *condition-by-deviance* interaction as fixed effects and participant-wise random intercepts (*m2*). We compared this model with models that contained only an intercept as a predictor (*m0*) and an intercept and condition (*m1*) as predictors, besides the participant-wise random intercepts. The model *m2* performed significantly better than both m0 (Chi$^2$(2) = 81.28, $p < 0.0001$) and m1 (Chi$^2$(2) = 41.76, $p < 0.0001$); AIC$_{m0}$ = 707.1, AIC$_{m1}$ = 671.6, AIC$_{m2}$ = 635.8. ANOVA on *m2* revealed significant effects of condition (F(2,170) = 31.38, $p < 0.0001$), deviance (F(1,170) = 37.15, $p < 0.0001$) and a significant condition-by-deviance interaction (F(2,170) = 7.38, $p = 0.001$). Post-hoc contrasts revealed that P2 amplitudes were significantly higher for standards than for deviants in the harmonic (contrast estimate = −1.12, SE = 0.19, $t$(170) = −5.79, $p < 0.0001$) and inharmonic (contrast estimate = −.83, SE = 0.19, $t$(170) = −4.262, $p = 0.0005$) conditions, but not in the changing condition (contrast estimate = −0.01, SE = 0.19, $t$(170) = −0.51, $p = 0.99$) (Table S5).

Taken together, these results suggest the presence of a frontocentral negativity in the *harmonic* vs. *inharmonic/changing* difference waves within a latency range of 100–200 ms (Fig. 5). This pattern was found for all studied comparisons except for the *harmonic vs. changing* deviants, likely due to the absence of a mismatch response in the latter (Fig. 5D). Note that the

inharmonic deviant had a more negative amplitude than the harmonic standard (contrast estimate = −2.06, SE = 0.19, $t$(170) = −10.61, $p < 0.0001$, see Table 1), suggesting the presence of inharmonicity effects in the absence of deviance effects.

## Behavioral experiment

To investigate if the observed responses were related to perceiving multiple auditory objects in both inharmonic conditions, we ran a follow-up behavioral study. The participants were asked to listen to short sequences of either *harmonic*, *inharmonic* or *changing* sounds and judge if they heard one, two or three or more sounds at once. Results revealed that listeners were over sixteen times more likely to judge sounds as consisting of many different objects in the *inharmonic* condition (OR = 16.44, Est. = 2.80, SE = 0.57, $p < 0.0001$) and over 62 times more likely in the *changing* condition (OR = 62.80, Est. = 4.14, SE = 0.64, $p < .0001$) in comparison to the *harmonic* condition (Fig. 6A).

In the behavioral experiment, we also took the opportunity to examine if the participants were able to consciously perceive the F0 deviants in the *changing* condition. To this end, we exposed the participants to sequences of 20 sounds with roving F0 (following the same rules as in the EEG experiment) and asked them to count the number of deviants. For each sequence, we calculated a counting error metric. We constructed a linear mixed model to predict these errors in this task with intercept and condition as fixed effects and per-participant random intercepts (*m1*). This model performed

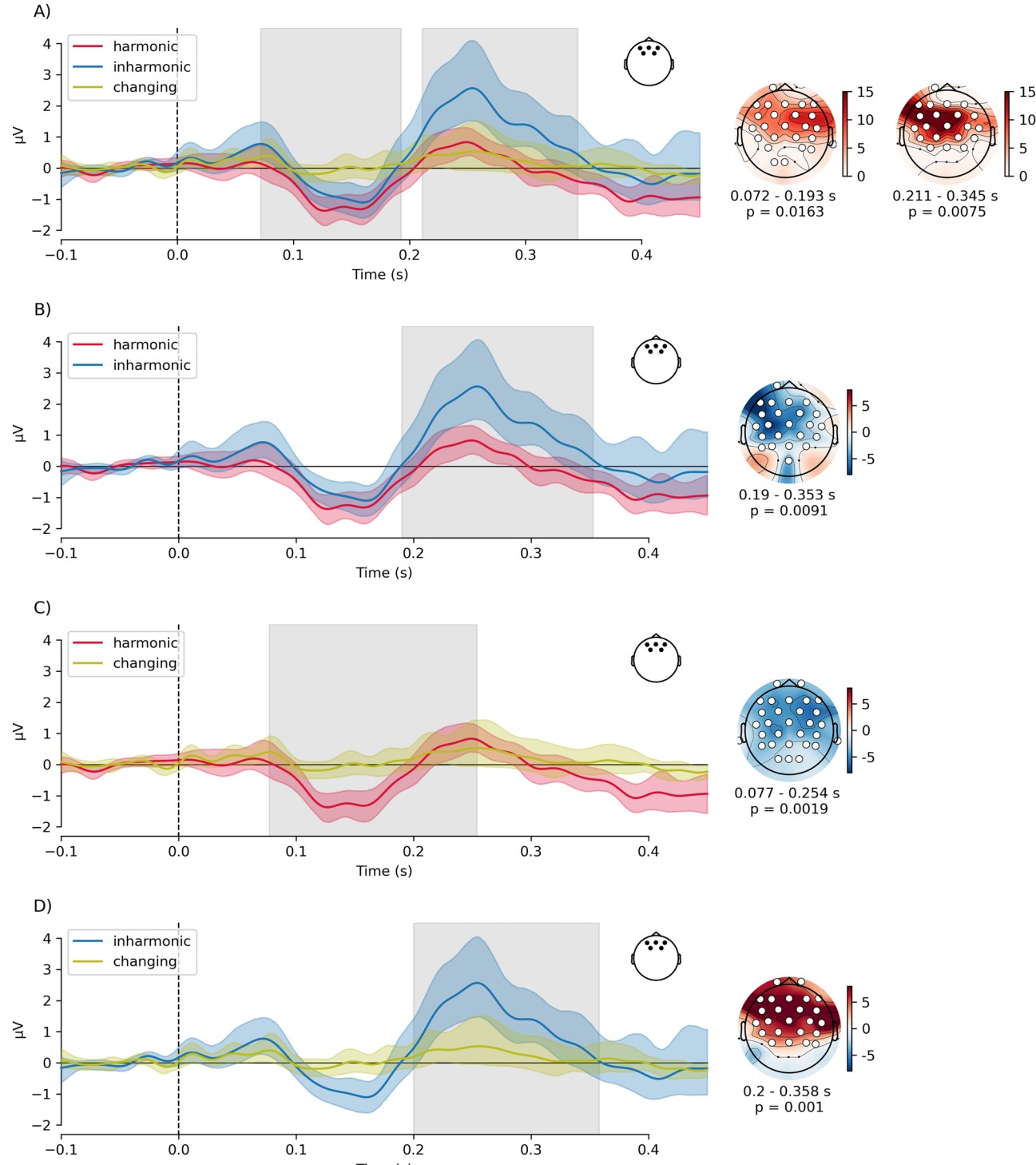

**Fig. 3 | Event-related potentials in the harmonic, inharmonic and changing conditions.** Traces show grand-average mismatch waves (deviant—standard, corresponding to green mismatch traces in Fig. 2) for the frontal channel activity (electrodes Fz, F3, F4, FC1, FC2). **A** compares responses between the three conditions, while **B–D** show post-hoc pairwise comparisons. Colored bands around the traces represent 95% confidence intervals around the mean. Gray rectangles indicate statistically significant clusters in the mass-univariate analysis. Topographies show F-maps (**A**) or t-maps for post-hoc comparisons (**B–D**) for each significant cluster, with the temporal extent of the clusters indicated below each topography. White markers represent channels that comprise the cluster.

significantly better than a null model ($m0$) that contained only intercept as predictor and per-participant random effect ($Chi^2(2) = 69.12$, $p < .0001$); $AIC_{m0} = 1203.3$, $AIC_{m1} = 1138.2$. Estimated marginal means for the absolute errors in the *harmonic* ($M = 0.31$, $SE = 0.40$) and *inharmonic* ($M = 0.62$, $SE = 0.41$) condition were not significantly different ($t(206) = -0.630$, $p = 0.80$). However, the absolute error in the *changing* condition ($M = 4.25$, $SE = 0.41$) was significantly higher than in both *harmonic* ($t(206) = 8.08$, $p < .0001$) and *inharmonic* ($t(206) = 7.42$, $p < .0001$) conditions. There was no significant difference between *harmonic* and *inharmonic* conditions. These results suggest that while participants were able to generate a predictive model and consciously perceive deviations from it in both *harmonic* and *inharmonic* conditions, this may not have been the case in the *changing*

**Fig. 4 | Peak measures differences in harmonic, inharmonic and changing conditions.** Panels show MMN mean amplitude (**A**), P3a mean amplitude (**B**), MMN peak latency (**C**), and P3a peak latency (**D**). Violin plots show distributions of obtained results. White dots represent the median, narrow rectangles represent quartile 2 and 3 ranges, and vertical lines represent minima and maxima. Gray lines connect observations from the same participant. Stars (***) and red bars represent statistically significant post-hoc comparisons (all *p*-values < 0.001).

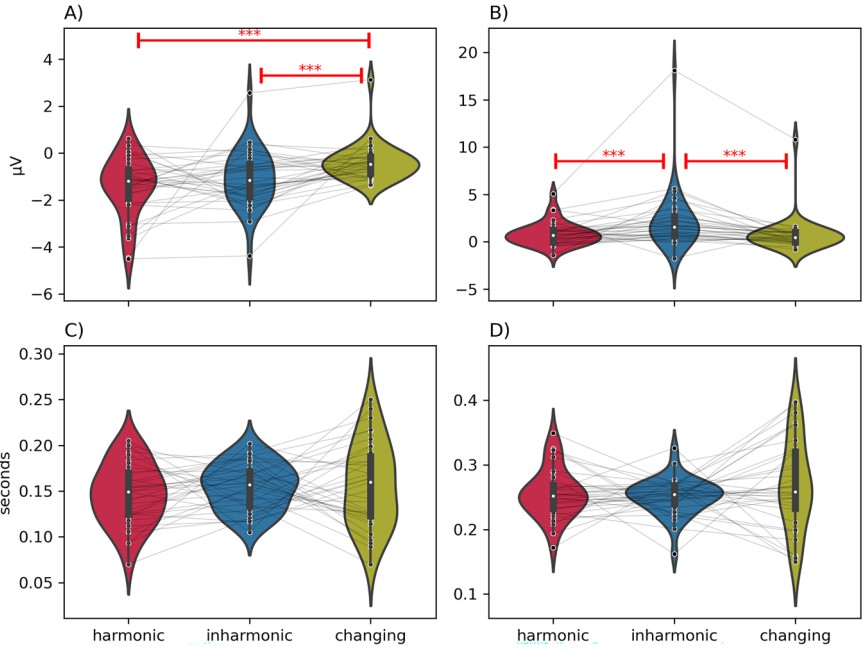

condition, where a predictive model did not seem to be formed, and many sounds were considered deviants.

### Second- and third-order deviants

Next, we took advantage of the roving paradigm to examine whether the effects of harmonicity continue for sounds after the first deviant. To this end, we analyzed mismatch responses to second and third sounds after the frequency change (the second- and third-order deviants in each "rove"). We applied the same logic as in the main analysis, first using a mass-univariate approach and then focusing on peak measures. In the mass-univariate comparison of standard and deviant responses, we only found significant differences for the second-order deviant in the inharmonic condition at the latency range of P3a at 215–450 ms (19/30 sensors, *p* = 0.0042), but nothing at the latency range of MMN. No statistically significant differences were found in the harmonic and changing conditions. Similarly, no significant differences appeared for the third-order deviant in any condition (*p* > 0.05 for all clusters). This indicates that the P3a response for inharmonic sounds continued to be detectable in the second-order (but not third-order) deviants, while there was no detectable MMN for these extra deviants. However, when we compared the three conditions using a cluster-based permutations *F*-test, we found no significant differences for second- as well as third-order deviants (*p* > 0.05 for all clusters, Table S3).

We investigated this result further using linear mixed modeling for mean amplitude and peak latency, with participant-wise random intercepts. For the second-order deviant, we found that the model *m1*, which predicted P3a mean amplitude with condition, performed significantly better than the null model *m0* ($AIC_{m0}$ = 344.6, $AIC_{m1}$ = 340.4, $Chi^2(2)$ = 8.21, *p* = 0.016). Post-hoc comparisons revealed significant differences between harmonic and changing (contrast estimate = 0.67, SE = 0.26, *t*(66) = 2.55, *p* = 0.035), as well as inharmonic and changing conditions (contrast estimate = 0.081, SE = 0.26, *t*(66) = 3.07, *p* = 0.009). No significant differences were found for the harmonic–inharmonic contrast. In the case of third-order deviants, the condition did not improve model performance for any of the studied peak measures (*p* > 0.05 for all model comparisons). These results suggest that the P3a response gets carried over to the second-order deviant and is stronger for both harmonic and inharmonic sounds than for changing sounds.

### Frequency shifts

Finally, we investigated whether the effects that have been observed thus far are moderated by the amount of change (deviance) of the F0. In the roving

paradigm, the F0 changes randomly in 50 Hz increments from 50 Hz to 300 Hz, both up and down. We extracted the amount of F0 change associated with each first-order deviant and entered it as a fixed effect into the linear mixed models that were analyzed previously. Models containing the main effects of condition and frequency shift (*m2*) performed significantly better than models that included just the condition (*m1*) for MMN mean amplitude ($AIC_{m1}$ = 3470, $AIC_{m2}$ = 3467, $Chi^2(2)$ = 5.15, *p* = 0.023) and P3 mean amplitude ($AIC_{m1}$ = 3712, $AIC_{m2}$ = 3703, $Chi^2(2)$ = 11.33, *p* = 0.001). However, models containing the interaction of condition and frequency shift (*m3*) did not perform significantly better than *m2* for all studied measures (all *p*-values > .05, see Supplementary Table 4 and Supplementary Fig. 2). These results indicate that the effects of harmonicity established in the previous analyses are not moderated by the amount of shift in the fundamental frequency.

### Discussion

In this study, we showed that harmonicity influences the brain's mismatch responses to both expected and unexpected sounds. Contrary to our hypothesis, inharmonic sounds with a constant jittering pattern (the *inharmonic* condition) generate MMN responses comparable to those of harmonic sounds, and elicit P3a responses that are stronger than in the other two conditions (despite the passive listening nature of the task). In contrast, MMN responses become undetectable when the jittering changes between sounds (the *changing* condition), suggesting that sequential, but not spectral uncertainty, induces the precision-weighting effect. Interestingly, the ERPs to both standards and deviants differed between harmonic and inharmonic sounds, suggesting that inharmonicity elicits an ORN response. This result can be further explained by behavioral data, suggesting that for inharmonic sounds, listeners are more likely to perceive more than one auditory object at the same time. Overall, our results suggest that inharmonicity does not act as a source of uncertainty as conceived by classic predictive processing theories. Instead, inharmonicity seems to induce the segregation of the auditory scene into different streams, capturing attention (as reflected in the P3a) and giving rise to specific neural processes that are independent from the predictive mechanisms underlying sequential deviance detection and the MMN.

The MMN in the *inharmonic* condition did not differ significantly from the *harmonic* condition. This result is not consistent with the precision-weighting hypothesis, as inharmonic sounds carry more information and should theoretically yield predictions of lower precision[38].

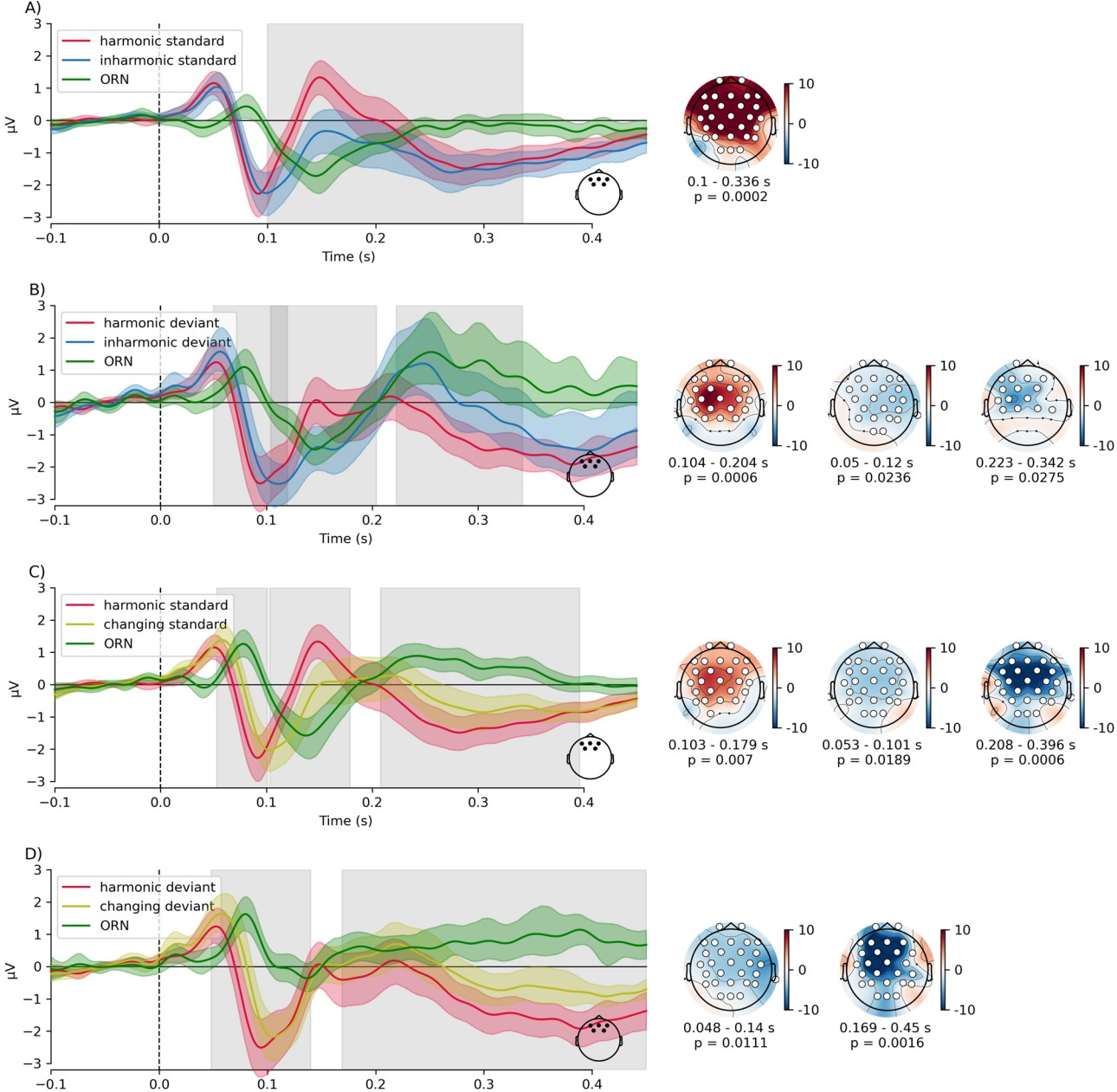

**Fig. 5 | Event-related potentials comparison for object-related negativity (ORN) analysis.** Traces show grand-average responses for harmonic, inharmonic and changing standards or deviants. Green trace shows the difference wave corresponding to the ORN. **A** compares harmonic and inharmonic standards, **B** compares harmonic and inharmonic deviants, **C** compares harmonic and changing standards, while **D** compares harmonic and changing deviants. Traces represent averaged frontal channel activity (electrodes Fz, F3, F4, FC1, FC2). Colored bands around the traces represent 95% confidence intervals around the mean. Gray rectangles indicate statistically significant clusters in the mass-univariate analysis. Topographies show t-maps for each significant cluster. White markers represent channels that comprise the cluster.

Instead, it suggests that the MMN is sensitive to the uncertainty present in the sequence of consecutive stimuli. This is evidenced by the fact that the MMN was undetectable in the *changing* condition, where spectral uncertainty introduced by inharmonicity was coupled with sequential uncertainty introduced by random jittering of consecutive sounds. In the *inharmonic* condition, all partials in the complex tone changed with every deviation; however, the relationship between the frequencies remained the same. This would mean that the MMN is sensitive to more global (context-dependent) uncertainty and not to the uncertainty generated by the introduction of inharmonicity.

This result relates to a larger issue of how precision-weighting is related to MMN, an ERP that is thought to reflect prediction error responses[25]

(Garrido et al., 2009). In general, precision-modulated evoked responses to unexpected stimuli could theoretically exhibit both larger and smaller amplitudes. A smaller ERP amplitude could result from a smaller predicted difference between the standard and a deviant (smaller prediction error). However, it could also result from a larger predicted difference that is down-weighted by precision[13,39]. Recent simulation work has shown that these two cases could in practice be disassociated, because any change to precision-weighting would necessarily be accompanied by a change in the latency of the ERP peaks[40]. We have not found any evidence of latency effects on any of the studied components.

The results of this study show that inharmonic sounds with *changing* jitter patterns produce weaker mismatch responses than harmonic sounds.

**Table 1 | Cluster-based permutations on harmonic vs. inharmonic and harmonic vs. changing contrasts**

| Contrast | Latency (ms) | Polarity | Sensors | *p*-value |
|---|---|---|---|---|
| Harmonic vs. inharmonic standards | 100–336 | Negative | 28/30 | 0.0003 |
| Harmonic vs. inharmonic deviants | 50–120 | Positive | 23/30 | 0.021 |
| | 104–204 | Negative | 23/30 | 0.008 |
| | 223–342 | Positive | 15/30 | 0.025 |
| Harmonic vs. changing standards | 53–101 | Positive | 29/30 | 0.023 |
| | 103–179 | Negative | 26/30 | 0.001 |
| | 208–396 | Positive | 29/30 | 0.001 |
| Harmonic vs. changing deviants | 48–140 | Positive | 27/30 | 0.013 |
| | 169–450 | Positive | 26/30 | 0.002 |

Latencies indicate the temporal span of detected clusters, while Sensors indicate how many channels contributed to each cluster. Note that only clusters with *p* < 0.05 are shown here. For a complete list of clusters detected in this analysis, see Table S2.

**Fig. 6 | Behavioral experiment results. A** shows the number of responses of each category between the three conditions. **B** shows the mean errors in the three conditions. These errors were calculated as the absolute value of the difference between participant-reported and ground truth number of deviants (lower values = more correct answers). Ground truth values ranged from 1 to 5 (*M* = 2.8, SD = 0.72). Dots represent single observations. Horizontal lines represent the median, boxes represent quartile 2 and 3 ranges, whiskers represent minima and maxima.

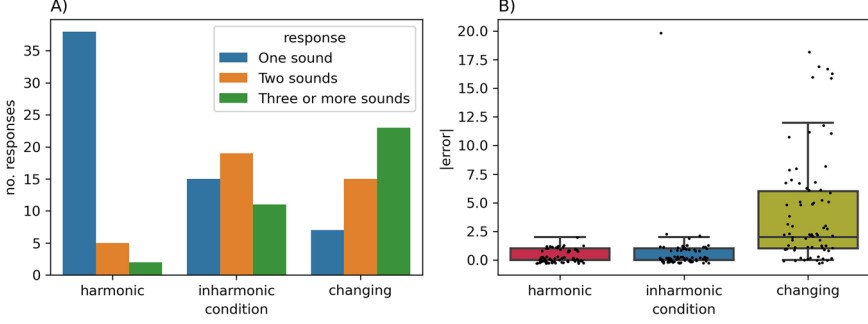

This is consistent with the precision-weighting hypothesis, as the *changing* condition has higher sequential uncertainty, making the inference about the pitch of each consecutive sound more uncertain (however, see *P3a, ORN and multiple auditory objects* section for an alternative explanation). This is also in line with previous research suggesting a role of harmonicity in precision-weighting for auditory features such as timbre, intensity and location[30]. However, the stimuli used in that study were sounds of musical instruments (piano and hi-hat) that differed in other acoustical properties apart from harmonicity (e.g., there were substantial differences in the spectra and in the amplitude envelopes). Furthermore, that study could not investigate pitch differences per se, due to the pitch percept being imperceivable for the hi-hat sound. Here, we provide a systematic manipulation of harmonicity using synthetic sounds, ensuring that all other acoustic parameters apart from harmonicity of the stimuli remain the same between the conditions. The robustness of these results is further reinforced by the lack of any interaction between frequency shift and condition, which indicates that the observed effects are not frequency-dependent.

Surprisingly, we found that P3a amplitude was higher for inharmonic than for harmonic sounds. Following the conventional interpretation of the P3a[29], this result may indicate that inharmonic sounds capture attention more than harmonic sounds. A possible explanation might be that the auditory system treats inharmonic sounds as multiple, different auditory objects, while it collapses harmonic sounds into a single prediction about the F0. The F0 change associated with the deviant in the *inharmonic* condition would then require an update to not one but many different predictive models, resulting in overall stronger prediction error responses. In contrast, in the case of the changing condition where sounds constantly shift, the auditory system attempts (and fails) to track changes in too many partials at once, because the only stationary frequency component is the F0. This logic could explain the P3a results in the present study, as well as the absence of both P3a and MMN components in the *changing* condition.

An unexpected result came from direct comparisons between responses to harmonic and inharmonic sounds (both standard or deviant).

The introduction of inharmonicity produced a response pattern characterized by a more negative P2 peak. We interpret this as an ORN, an ERP related to the separation of concurrently presented sounds into distinct auditory objects[35–37]. The ORN is thought to reflect the auditory system performing concurrent sound segregation. Conventionally, it is elicited in paradigms with complex tones that include one mistuned harmonic or a harmonic with an asynchronous onset[36,41]. Other ORN-eliciting cues include dichotic pitch[42], simulated echo[43], differences or discontinuities in location[44], and onset asynchrony[45]. Importantly, the ORN is not a mismatch response, i.e., it does not arise in response to a violation of any global rule established with an oddball paradigm. This fact enabled us to evaluate the ORN using standard vs. standard and deviant vs. deviant comparisons. Importantly, the ORN is thought to arise from different neural sources than the MMN[46]; however, we did not see topographic differences in our data (see Supplementary Fig. 3). We found that it was elicited by both inharmonic standards and inharmonic deviants. One explanation of this result is that the auditory system interprets the inharmonic sound as multiple different sound sources and performs auditory object segregation. This is independent of pitch deviance detection indexed by the MMN. This conclusion is also reinforced by the presence of ORN in the harmonic standard vs. the changing standard comparison. Finally, the P3a results discussed above also support this conclusion.

When synthesizing sounds for this experiment, we used every frequency in the harmonic series up to the Nyquist limit at 24 kHz. This approach was motivated by the need to ensure that the harmonicity manipulation is applied equally throughout the frequency spectrum and that the F0 remains clearly audible throughout the sequence. We rejected frequency-jittering patterns that could produce beating artifacts arising from two harmonics being too close to each other in frequency. Thus, our stimuli were broadband, spanning a large portion of the human hearing range. However, in psychophysical studies on pitch perception, the stimuli are often high-pass or band-pass filtered specifically to reduce the amplitudes of the low frequency harmonics[8,47]. This procedure makes the pitch perception system focus on the so-called "temporal fine structure" (the

high-frequency content) of the sound and does not rely on the resolvability of the lower harmonics[48]. This issue can be addressed in the future by experimentally varying the amount of low-frequency information available.

A related issue is the fact that the lack of a significant MMN in the changing condition can be (at least in part) explained by the amount of jittering applied to the harmonics. We used jittering rates sampled from a uniform distribution in the range between -0.5 and 0.5, which corresponds to up to 50% change in the frequency. This change is quite salient perceptually and leads to a situation where the frequency deviants in the *changing* condition were not discernible from the continuously changing standards, as evidenced by the behavioral results. This value was chosen because it was used in a series of behavioral studies that established effects of harmonicity on pitch-related and memory tasks[8,9,49]. However, it is possible that with lower jitter values, the MMN and P3a components would be apparent in the *changing* condition as well. Thus, the relationship between the strength of mismatch responses and jitter rate should be thoroughly addressed in future research. Relatedly, more research is needed to establish the behavioral thresholds of pitch detection as a function of jitter rate.

It is important to note that any precision-weighting mechanisms in the brain can only make use of information that is already encoded by the peripheral sensory system. In this sense, a more valid metric of precision would be the entropy associated with the output from the cochlea. While any cochlear transformation would necessarily retain many aspects of the harmonic structure of sound, we acknowledge that the correlation between the entropy of the acoustical wave and the entropy of its neural representation is not perfect. Future research should address this issue, perhaps by using computational models of the cochlea and the auditory nerve.

In sum, this work suggests that inharmonicity does not serve as an index of precision-weighting for low-level, short-timescale auditory predictions, as evidenced by the MMN results. Instead, it encourages the auditory system to perform object segregation, as evidenced by the ORN results. The tracking of multiple objects (or multiple predictions) engages attention and leads to larger P3a responses to unexpected changes as long as multiple objects are trackable within a sequence (as in the *inharmonic* condition). This work offers a new perspective on the neural mechanisms underlying the auditory processing of low-level acoustic features, as well as expanding previously established behavioral effects of inharmonicity on pitch perception and memory.

## Methods
### Participants
We recruited 37 participants for the EEG experiment. One participant completed the study but was removed from analysis due to audio equipment failure during the procedure. Another participant was removed from analysis because of high levels of noise present in the data (over 60% of epochs excluded in the *autoreject* procedure, see below). The final sample size was 35 participants (median age 27 years, 18 female). We included participants with no reported neurological or psychiatric illness, aged between 18 and 45 years old, with normal hearing, normal sight or corrected normal sight (e.g., contact lenses) and no use of medication that affects the central nervous system (e.g., opioids, pain medications). All participants received monetary compensation for their participation and signed an informed consent form. The study was approved by the department's Institutional Review Board at Aarhus University (reference number DNC-IRB-2022-009) and followed the Declaration of Helsinki. All ethical regulations relevant to human research participants were followed.

The number of participants was calculated with power analysis before the experiment. We used linear mixed models on a simulated dataset with the hypothetical differences between harmonic and inharmonic conditions (see *Stimuli and procedure*) set at 1 µV. This choice was informed by the results of a previous study where the difference in peak amplitude between harmonic piano and inharmonic hi-hat tones was around 1µV[50]. Here, we aimed to be able to detect a similar difference between harmonic and inharmonic conditions, with an assumption that *changing* would not differ significantly from the *inharmonic* condition (see *Procedure* for the

descriptions of the conditions). We ran this simulation 10,000 times for each $N$ value in the range between 25 and 40. The analysis showed that in order to achieve a statistical power of 0.8, at least $N = 33$ participants were required.

For the behavioral study, 15 participants were recruited (median age 20 years, 13 female). Inclusion criteria were identical to the ones used in the EEG experiment. In this case, the participants did not receive monetary compensation.

### Stimuli and procedure
The stimuli consisted of complex tones made by adding sine waves of varying fundamental and harmonic frequencies up to the Nyquist limit (24 kHz). Harmonics were added with equal amplitudes, in sine phase. Each tone had a duration of 70 ms and was amplitude modulated with 5 ms onset and offset ramps. To synthesize inharmonic stimuli, a procedure was adapted from McPherson and McDermott (2018). Each frequency in the harmonic series above F0 was jittered by a random value to introduce inharmonicity. The jitter value was calculated by multiplying the original frequency by a number drawn from a uniform distribution $U(-0.5, 0.5)$. In order to avoid beating artifacts, rejection sampling was used in cases where two frequencies would be spaced closer than 30 Hz apart. In the inharmonic condition, the same jitter pattern was applied to all sounds in the stimulus block. In the inharmonic-changing condition, each stimulus train had a new, randomly generated jitter pattern (Fig. 1). A sound pool of one harmonic and one thousand inharmonic sounds for each F0 was generated offline before the study as 16-bit .wav files with a sample rate of 48 kHz. All sounds were loudness-normalized to −12 LUFS using *pyloudnorm*.

Tones were presented using an oddball roving paradigm with an inter-stimulus interval of 600 ms. Within each stimulus train, all standard tones had the same F0 and were followed by a train of stimuli with different F0 (chosen randomly from a 500 –800 Hz range in 50 Hz intervals). In this paradigm, the first tone of a new stimulus train serves as the deviant stimulus and a potential source of mismatch negativity in the context of prior stimuli. After a few repetitions, this deviant tone is established as the new standard until the next stimulus train is presented. The number of stimuli in a given train varied pseudo-randomly from 3 to 11, with 3 to 7 repetitions being four times more probable than 8 to 11. The entire procedure was divided into six blocks (two for each harmonicity condition), each 6 min long. Each block consisted of 600 sounds, with 98 deviants on average. The participants were asked not to move during the auditory stimulation; however, there were short breaks between the blocks to provide rest and relaxation. All stimuli were administered passively, and participants were watching a silent movie throughout the procedure. The stimuli were randomized and played back using PsychoPy (version 2022.1.3). Headphones (Beyerdynamic DT 770 PRO) were used for binaural sound presentation. The entire experiment lasted 80-90 minutes on average, including participant preparation and debriefing.

The behavioral experiment consisted of two parts. In the first part, participants listened to a sequence of 8 sounds at a stationary F0 and were asked if they perceived "one", "two", or "three or more" sounds "at the same time, at any given moment". Depending on the condition, the sequences could be either *harmonic*, *inharmonic* or *changing*, following the principles from the EEG experiment and using the exact same sound pool. Three sequences were presented for each condition for a total of nine trials per participant.

In the second part of the behavioral task, participants listened to a sequence of 20 sounds with varying fundamental frequencies (F0s) that followed the rules of the EEG experiment. The task was to silently count the number of deviants ("please count, how many times the sounds change their pitch") and report the number after each sequence. This was preceded by a demo trial that included a harmonic sequence and informed the participant about the number of deviants in that sequence. The participants could repeat the demo trial multiple times until they became familiar with the task. Overall, five sequences were presented for each condition (*harmonic*, *inharmonic* or *changing*) for a total of 15 trials per participant. The entire experiment lasted about 5-7 minutes on average and was performed on a

laptop running PsychoPy in a quiet room. The sounds were presented binaurally using Beyerdynamic DT 770 PRO headphones.

## Statistics and reproducibility

Scalp EEG potentials were recorded with a 32-channel active system (BrainProducts GMBH) at a sampling rate of 1000 Hz. All EEG signal processing was performed in Python v.3.11 using *MNE* v.1.5[51,52], *Numpy* v.1.24[53] and *Pandas* v.2.1[54]. *Matplotlib* v.3.7 was used for plotting[55]. Horizontal and vertical eye movements (EOG) as well as heart rate (ECG) were recorded with additional electrodes. Raw EEG data were high-pass filtered at 0.2 Hz, divided into epochs (from -100 ms to 450 ms) and entered into the *autoreject* algorithm v.0.4.2[56] for automatic bad channel and bad epoch selection. This method uses cross-validation and a robust evaluation metric to estimate optimal peak-to-peak thresholds for each EEG sensor. This thresholding is applied epoch-wise, and the signals are either interpolated or (in case of many bad sensors) rejected. Afterwards, an independent components analysis (ICA) was performed in order to remove eye movement and heart-related artifacts. ICA components were marked for removal using an automated procedure based on data from EOG and ECG channels (the functions *find_bads_eog()* and *find_bads_ecg()* from the *MNE* package). However, these automatic choices were inspected visually. A maximum of four ICA components were removed from any participant. The ICA solution was applied to the data before artifact rejection, and *autoreject* was performed once more, as suggested by Jas et al. [56]. Overall, 6.9% of epochs were rejected. Finally, the epoched data were low-pass filtered at 30 Hz, re-referenced to the mastoids and baseline-corrected with a baseline of 100 ms pre-stimulus.

We treated the sound as a deviant if it was the first, second or third sound after the change of F0 in the roving paradigm. All non-deviant sounds were treated as standard, apart from the first five sounds at the start of each experimental block. ERPs were estimated by averaging the responses to standard and deviant sounds across participants and across conditions. Difference waves were calculated by subtracting the standard from the deviant ERP, for each deviant type (first, second or third), each participant and condition. For ORN analysis, differences between harmonic and inharmonic or changing ERPs were calculated, separately for standards and (first-order) deviants. In the mass-univariate approach, entire ERPs were subject to cluster-based permutation analyses. In peak-based analyses, we extracted participant-wise latencies for MMN and P3 peaks in the fronto-central EEG channels. For MMN, we extracted the latency of the most negative peak between 70 ms and 250 ms. For P3a, we found the latency of the most positive peak between 150 and 400 ms. The mean amplitude of each component was calculated as the average signal amplitude in the 50-ms window centered at the peak.

Cluster-based permutations were performed using MNE's built-in functions. Thresholds were chosen automatically based on the number of valid observations and assuming a *p*-value of .05. For each test, 1024 permutations were performed. Linear mixed models were fitted in R v.4.4.1[57] using *lme4* v.1.1[58]. Model comparison was performed with a likelihood ratio Chi[2] test. Post-hoc comparisons were performed by comparing the estimated marginal means calculated with *emmeans* v.1.10[59]. *p*-values were corrected for multiple comparisons using the Tukey HSD method. Approximate entropies were calculated with *AntroPy* v.0.1.6[60]. Entropy was estimated and averaged for all inharmonic sounds within the sound pool.

The behavioral experiment was performed on a laptop computer running PsychoPy. The sounds were presented binaurally using Beyerdynamic DT 770 PRO headphones. Participants gave their responses by pressing the keyboard. Count error metrics were calculated by taking the absolute value of the difference between participant-reported and ground truth number of deviants. A cumulative link mixed model regression with participant-wise random intercepts was used to estimate odds ratios in the first task. A linear mixed model was fit in the second task.

## Reporting summary

Further information on research design is available in the Nature Portfolio Reporting Summary linked to this article.

## Data availability

The raw data from this experiment were stored in a Zenodo repository and is freely available at https://doi.org/10.5281/zenodo.13939896.

## Code availability

The code that was used to perform this experiment and analyze the results is freely available at https://doi.org/10.5281/zenodo.15236581.

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

## Acknowledgements
We would like to thank Thies Lübeck, Pelle de Deckere and Marc Barcelos for their help in data collection and data preprocessing. We would also like to thank Alejandro Blenkmann and Andrew Oxenham for inspiring discussions that led to some of the ideas presented in this paper. Krzysztof Basiński is supported by grants from the National Agency for Academic Exchange, Poland (Bekker, no. BPN/BEK/2021/1/00358/U/00001) and National Science Center, Poland (Sonata, no. 2022/47/D/HS6/03323). Alexandre Celma-Miralles and Peter Vuust are funded by the Danish National Research Foundation (DNRF117) supporting the Center for Music in the Brain. David R. Quiroga-Martinez is supported by the Carlsberg Foundation (CF23-1491).

## Author contributions
K.B.: conceptualization, methodology, software, data analysis, investigation, data curation, funding acquisition, writing—original draft, review & editing, visualization. A.C.M.: conceptualization, methodology, investigation, writing—review & editing. D.R.Q.M.: conceptualization, writing—review & editing. P.V.: funding acquisition, supervision, writing—review & editing.

## Competing interests
The authors declare no competing interests.
