## [Transparent Peer Review file · Communications Biology]

Inharmonicity enhances brain signals of attentional capture and auditory stream segregation

Corresponding Author: Dr Krzysztof Basiński

Version 0:

Reviewer comments:

Reviewer #1

(Remarks to the Author)

This study explores how changes in sound structure affect MMN and P3 brain responses during listening. In particular, the researchers investigated the effect of spectral uncertainty on these ERPs by comparing harmonic, inharmonic, and changing sound contexts. They found that harmonic and inharmonic contexts elicited similar MMNs, with inharmonic contexts generating stronger subsequent P3 responses. Changing sound context did not show any clear MMN response. Finally, inharmonic contexts also elicited Object-related Negativity. The authors suggest that less structured sounds induce the brain to separate them into different streams and engage processes beyond the predictive mechanisms underlying deviance detection.

Overall, this is an interesting study. Here are a few issues that still require attention.

- 1) The ORN analysis does not entirely convince me. Specifically, I cannot understand why the contrast between unsubtracted signals (e.g., standard vs. standard, or deviant vs. deviant) should indicate auditory segregation. Please clarify.
- 2) The authors rightly report that many sounds in nature are harmonic (see lines 31-33 at the beginning of the Introduction). This consideration suggests that harmonic sounds may be more frequent than inharmonic ones, particularly among auditory signals relevant to the individual, indicating that the brain may be more trained to perceive harmonic patterns. Could this greater familiarity with harmonic contexts explain (predictively or not) the results presented here?
- 3) EEG signal processing: (a) Lines 575-576: Which parameters were chosen to select whether either a channel or epoch was bad? (b) 577 – 578: Which parameters (algorithm) were used to define whether an ICA component was artifactual? (c) How many ICA components were rejected? (d) How was canonical artifact rejection carried out?
- 4) Line 59: "Here, we hypothesize that harmonicity might drive precision-weighting of prediction errors..." Stated this way, it appears that harmonicity is the only relevant perceptual (acoustic) feature; however, we know that the MMN is also elicited by violations of other features (e.g., duration, intensity, etc.).
- 5) Figure 1: I would probably group the plots belonging to the two main conditions (i.e., A, E, and F; and B, G, H).

Reviewer #3

(Remarks to the Author)

Brief summary of the manuscript

This is a study of the extent to which uncertainty in the spectral domain affects markers of auditory prediction-error signals. Electrophysiological measures (MMN and P3a) during a passive-listening roving oddball paradigm were not diminished in amplitude for inharmonic compared to harmonic sequences. However the introduction of sequential uncertainty (different jitter of harmonic components for successive complexes, the "changing" condition) led to elimination of the MMN. A different electrophysiological measure resembling the object-related negativity was detected in inharmonic and changing conditions. The authors discuss the results in terms of relationships between processes of scene segregation, pitch perception, and

deviance detection.

Overall impression

The experimental design is appropriate with well controlled stimulus manipulations. Behavioural experiments shed further insight on the electrophysiological data. The manuscript is mostly very clear with appropriate description of methods and visualisation of data. From the outset I was unsure about the premise of the hypothesis, i.e. that the spectral information content of a single (complex) stimulus (as opposed to a distribution of successive stimuli) can be considered a relevant form of precision to influence predictions/error processing relating to future stimuli. As such I was not too surprised by the core result. The findings are nonetheless interesting and worth reporting, particularly given the post-hoc but coherent alternative explanation for the electrophysiological findings.

Specific points

- 1) 56-57 - There is in fact a lot of work on precision-weighting and audition, including as cited at 70-72. The hypothesis outlined at 59-65 is indeed new, to my knowledge, but the sentence at 56-57 understates what has been done.
- 2) 59-65 - It would help to elaborate (here or maybe in discussion) why the instantaneous entropy of one feature of a single stimulus (rather than that of a distribution, as is manipulated) should affect future predictions (and weighting of errors relating to them). Is the rationale in this case that inharmonic sounds have a less clear pitch percept and it is therefore harder to make predictions about the pitch of future sounds than in the harmonic case? Is it true that the only relevant predictions are those relating to overall pitch, rather than those relating to the frequency of individual components? I guess this last point is addressed in the discussion, but it is a question I had on reading the hypothesis and might be grappled with up front.
- 3) Fig. 1. Somewhat related to the previous point, components have been coloured as standards/deviants based on whether they are part of a complex with the same F0 (perhaps better described as "lowest frequency" if there is no pitch percept). This seems to reflect an assumption that pitch is the relevant feature being tracked. However looking at 1C it stands out as incongruous that some bars are grey (i.e. standards) even if there was no preceding component with a similar frequency. The choice to consider overall pitch (rather than frequencies of components) as the feature of interest should be explicitly mentioned.
- 4) 104-105 - "the distance between consecutive partials changes between B and C" is unclear. I think the point is that the distance is fixed in B, which is different from C where the distances change. It is also not obvious what "consecutive partials" means, especially in 1C where it is hard to see which components are linked to which (does consecutive refer to the temporal or spectral dimension?)
- 5) 116-117 - it is not clear whether this is generically about the roving paradigm (use of present tense) or the roving paradigm in this example (use of simple past tense)? The passage "...sounds are presented at a specific frequency" does not apply in this experiment with complex sounds.
- 6) 146 - Is "signal strength" averaged over all tones and timepoints? Or is it in relation to the peak of the difference wave, as suggested at 155?
- 7) Fig. 3 - You could help the reader by reiterating that these plots are the same as green curves in Fig. 2
- 8) Fig. 3 - It would help to label the topographies with the time windows so the reader immediately sees why there are two in the first row (i.e. that they correspond to time windows not e.g. different pairs of conditions)
- 9) 196 - What was the rationale for the stricter p-value here than elsewhere?
- 10) 202-203 - mentions only one null model but the description suggests there are two - the text doesn't quite scan, perhaps has been incompletely edited from a previous version
- 11) 277-282 - I found this initially hard to follow - it doesn't seem to lead on from the previous paragraph which was about standard vs. deviant comparisons, not condition comparisons. Does "difference waves" here refer to conditions rather than standard vs. deviance? It might help the reader to refer back to particular figures .
- 12) 294-310 - It would be helpful to know how many deviants there were to interpret the scale of error (without having to turn to the methods)
- 13) Fig. 6 - what do the diamonds represent?
- 14) 402 - Again here there is an assumption that MMN reflects something relating to the pitch of the complex rather than operating on components (although I see the alternative is discussed at 415-426).
- 15) 437 - A different source for the ORN vs other components is referred to. Can you say whether this is the case in your own data (Fig. 5)
- 16) 445 - I wondered whether it was possible to include "every frequency". I then saw the limit mentioned at 521 - it would be worth rephrasing here too.
- 17) 560 - Might the poorer behavioural deviance detection in the "changing" condition be due in part to the fact that training was done only on the harmonic condition?
- 18) 584-590 - Why were the first three sounds after a change considered deviants for the main analyses but only the first for the ORN analysis? How was the number 3 chosen (rather than e.g. 2 or 4 sounds after the change).

Style/typos

- 19) 52 - add comma after "context"
- 20) 60 - remove extra bracket
- 21) 111 - spectra -> spectrum
- 22) 143 better as "... responses to standards (blue) and deviants (red), and the difference waves (green) ..."
- 23) 248 - should refer to Figure 2 not Figure 3?
- 24) 480 - "encourages" may be better than "enables" (as the system presumably already has the capacity)
- 25) 484 - "expands"  "expanding"

- 26) 608 - "Behavioural"  "The behavioural"

Version 1:

Reviewer comments:

Reviewer #3

(Remarks to the Author)

My thanks to the authors for addressing all my comments, I recommend accepting the submission.
